# The influence of the brittle-ductile transition zone on aftershock and foreshock occurrence

Giuseppe Petrillo[1], Eugenio Lippiello [1]✉, François P. Landes [2] & Alberto Rosso[3]

Aftershock occurrence is characterized by scaling behaviors with quite universal exponents. At the same time, deviations from universality have been proposed as a tool to discriminate aftershocks from foreshocks. Here we show that the change in rheological behavior of the crust, from velocity weakening to velocity strengthening, represents a viable mechanism to explain statistical features of both aftershocks and foreshocks. More precisely, we present a model of the seismic fault described as a velocity weakening elastic layer coupled to a velocity strengthening visco-elastic layer. We show that the statistical properties of aftershocks in instrumental catalogs are recovered at a quantitative level, quite independently of the value of model parameters. We also find that large earthquakes are often anticipated by a preparatory phase characterized by the occurrence of foreshocks. Their magnitude distribution is significantly flatter than the aftershock one, in agreement with recent results for forecasting tools based on foreshocks.

[1] Department of Mathematics and Physics, University of Campania "L. Vanvitelli", Viale Lincoln 5, Caserta 81100, Italy. [2] TAU, LRI, Univ. Paris-Sud, CNRS, INRIA, Université Paris-Saclay, Orsay 91405, France. [3] LPTMS, CNRS, Univ. Paris-Sud, Université Paris-Saclay, Orsay 91405, France. ✉email: eugenio.lippiello@unicampania.it

Earthquakes occur in brittle regions of the crust characterized by a velocity-weakening friction, which is at the origin of the stick-slip behavior. The distribution of friction along the fault plane is highly heterogeneous with strong spots, usually called asperities[1]. Asperities are expected to be surrounded by weak zones with a rheological behavior better described by a velocity-strengthening friction. When the stress accumulated in the surroundings of the hypocenter overcomes the local friction, an abrupt slip takes place and stress is redistributed in the surrounding regions. The stress redistribution along the brittle, velocity-weakening, part of the crust triggers the occurrence of other earthquakes, the aftershocks. They follow well established empirical laws that can be put in the form of power laws with quite universal values for the exponents[2]. In particular, the aftershock rate exhibits a roughly hyperbolic decay with time since the mainshock, an empirical law known as the Omori–Utsu law[3].

At the same time, the stress redistributed by the mainshock in velocity-strengthening regions induces some slow deformations, commonly defined as afterslip. Under the hypothesis[4,5] of proportionality between seismicity rate $\lambda(t)$ and afterslip rate, several features of aftershock occurrence are reproduced[4–10]. In ref. [11], we have demonstrated this proportionality in a model with only two elastically coupled degrees of freedom. The first described the fault displacement, with an heterogeneous velocity-weakening friction, while the second corresponded to the ductile region displacement, with a velocity-strengthening friction. This very simple description can model different tectonic contexts and suggests that the coupling with a velocity-strengthening layer and the heterogeneity in the fault friction are the two key ingredients controlling aftershock triggering. The same two ingredients are central in the pre-slip hypothesis[12–14] according to which small earthquakes, usually named foreshocks[15,16], are expected to anticipate the mainshock occurrence. According to this hypothesis, because of friction heterogeneity, there are small regions on the fault that have less resistive power than the large fault and can break before it, in presence of an underlying slow-deformation process. This mechanism can produce an increase of the seismic activity, as the occurrence time of the mainshock is approaching but, because of the limited number of foreshocks, it is very difficult to be identified[17–20]. Nevertheless, accurate investigations before some recent large earthquakes have elightened the presence of foreshocks together with a phase of slow slip of the plate

interface[21–23]. Other precursory patterns are observed if one considers the distribution in space of foreshocks[24–27] and/or their distribution in magnitude[28–31]. In particular, very recently, Gulia and Wiemer[31] have shown that the magnitude distribution during aftershock activity is steeper than during foreshock activity. This result is however achieved for only two mainshocks and by means of different selection criterions for the foreshock identification[32].

In this article, we show that friction heterogeneities and the slow deformation of a velocity-strengthening layer are sufficient ingredients to explain the whole ensemble of instrumental findings regarding the organization in time, space, and magnitude of both aftershocks and foreshocks. To this extent, we combine the model of two blocks of Lippiello et al.[11] with the description of the fault plane originally proposed by Burridge and Knopoff (BK)[33]: a two-dimensional elastic interface with many degrees of freedom, each being subject to a velocity-weakening friction law. Therefore our model of the fault consists in a collection of sliding blocks connected to a more ductile region, itself treated as an extended interface subject to velocity-strengthening rheology. This system has a clear geophysical justification and allows us to study the organization of simulated earthquakes not only over time but also in space and in magnitude. We find that the model reproduces the most relevant empirical laws observed for instrumental aftershocks and foreshocks, quite independently of the precise value of model parameters.

## Results and discussion

**The model.** The model we propose is composed by a first layer H that represents the brittle part of the fault. H is elastically coupled to a second layer U that mimics the ductile region below the fault and is driven by the tectonic dynamics at the (very small) velocity $V_0$. Each layer is an extended object made of many interacting degrees of freedom labeled $i = 1, 2, …, N$, organized on a square lattice. For simplicity, we assume a motion restricted along the $V_0$ direction, with scalar displacements $h_i(t)$ in the layer H and $u_i(t)$ in the layer U. In Fig. 1, we present a schematic description corresponding to a one-dimensional cut of the mechanical model along the $V_0$ direction. The model also extends in the other direction, which is orthogonal to $V_0$. From continuum mechanics, the elastic cost of the displacement field is $k_h \sum_{j \neq i}(h_j - h_i)/r_{ij}^2$, where $r_{ij}$ is the distance between points $i$ and $j$. The constitutive equations for the displacements $h_i$ in the layer H are obtained from the balance between the elastic forces and the velocity-weakening friction force $\tau_h$:

$$\tau_h = k_h \sum_{j \neq i} \frac{h_j - h_i}{r_{ij}^2} + k(u_i - h_i). \tag{1}$$

To improve the efficiency of our numerical scheme, we restrict the sum in Eq. (1) to nearest neighbors $|r_{ij}| = 1$, which corresponds to replacing the elastic force with the discrete Laplacian $k_h \nabla^2 h_i$. The total stress on $i$ simplifies to $k_h \nabla^2 h_i + k(u_i - h_i)$ (it is balanced by the friction $\tau_h$). We also apply this short-range approximation to the layer U, which is, however, more ductile. For this reason, we assume that the viscoelastic interactions[34] in U are implemented assuming that neighboring degrees of freedom are connected by means of a dashpot and a spring placed in series (Fig. 1)[35,36]. The constitutive equations for the layer U reads:

$$\tau_{u_i} = k_u(\nabla^2 u_i - z_i) + k(h_i - u_i) + k_0(V_0 t - u_i) \tag{2}$$

$$\eta \dot{z}_i = k_u(\nabla^2 u_i - z_i), \tag{3}$$

where $z_i$ is the viscoelastic degree of freedom, and the dot indicates a temporal derivative. The viscoelastic force $k_u(\nabla^2 u_i - z_i)$

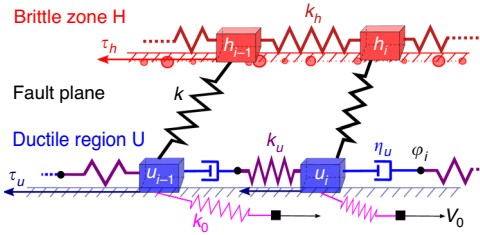

**Fig. 1 The mechanical model.** Mechanical sketch of the model (one-dimensional cut: the other direction is orthogonal to the plane). This is the direct extension of Fig. 1 from ref. [11]: each fault is modeled as a two-dimensional layer (and no longer as a single block). The fault plane $H$ is subject to velocity-weakening friction $\tau_h$, in the form of randomly placed pinning points (red disks) with varying pinning strength $\tau_i^{th}$ (disk radius). The ductile region $U$ is subject to velocity-strengthening friction $\tau_u$, and is pulled at constant velocity $V_O$ by distant regions. Within this ductile region, interactions are viscoelastic (Maxwell model), with dashpots having viscosity $\eta_u$ and elasticity $k_u$. The relative elongations of dashpots around site $i$ is denoted $z_i = \varphi_i - u_i - (\varphi_{i-1} - u_{i-1})$. The two layers are connected elastically with a stiffness $k$.

has an intrinsic timescale $t_\eta = \eta/k_u$. When $u_i$ moves, for times shorter than $t_\eta$ the dashpot variable $z_i$ remains frozen, so the term $k_u(\nabla^2 u_i - z_i)$ acts as a genuine elastic stress, and the layer U is solid-like. At longer times, the variable $z_i(t)$ evolves to suppress the viscoelastic force ($z_i = \nabla^2 u_i$), and the layer U displays a liquid-like behavior.

Finally, we have to define the form of the friction forces. For the force $\tau_u$ of the ductile layer U, we assume a velocity-strengthening friction, taking the stationary form of the rate-and-state friction (RSF) law[37–39]:

$$\tau_u(t) = \sigma_N \left( \mu_c + A \log \frac{\dot{u}_i(t)}{V_c} \right), \qquad (4)$$

where $\sigma_N$ is the effective normal stress, $\mu_c$ is the friction coefficient when the block $U$ slides at the steady velocity $V_c$ and $A > 0$ for a velocity-strengthening material.

For the friction in the brittle fault $H$, a random Coulomb failure criterion is adopted. As soon as the force overcomes a local random frictional stress threshold $\tau_i^{th}$, the position $h_i$ becomes unstable and moves forward by a random amount $(\Delta h)_i$. Slips of this kind are the bulk of earthquakes and occur on the very fast timescale $t_s$, typically of the order of seconds. It is reasonable to assume that $t_s$ is the shortest timescale in the problem, and by far, $t_s \ll t_\eta$, i.e., we assume it is instantaneous. Thus during an earthquake the layer $U$ behaves elastically and Eq. (2) can be approximated by

$$\tau_{u_i} = k_u \nabla^2 u_i + k(h_i - u_i) + k_0(V_0 t - u_i) \qquad t \sim t_s \ll t_\eta, \quad (5)$$

the term $k_u z_i$ being constant at these timescales, it plays no role in the dynamics of $\tau_u$, $u_i$. As a consequence, for each slip $(\Delta h)_i$ at position $i$ in $H$, there are slips $(\Delta u)_j = q_{r_{ij}}(\Delta h)_i$ at all positions $j$ in the layer U, where $q_{r_{ij}}$ is a decreasing function of the distance $r_{ij}$. In general, the precise form of the $q_{r_{ij}}$ depends on the details of the dynamics of $h_i(t)$, $z_i(t)$, $0 < t < t_s$ and can be quite complicated. Indeed, when we apply the RSF laws combined with all other equations (Eq. (3) in particular) to compute the true form of $q_{r_{ij}}$, we find a very fast decay as a function of $r_{ij}$, and thus decide to neglect terms that are not nearest neighbor to the slipping site. Thus in practice we use a short-range form for $q_{r_{ij}}$: $q_{r_{ii}} = q_0$, $q_{r_{ij}} = q_1$, if $|r_{ij}| = 1$ and 0 for all others. After the earthquake, at times $t > t_\eta$, the dashpots of the layer U are relaxed and have dissipated some elastic stress (the $k_u \nabla^2 u_i$ term is exactly compensated by $-k_u z_i$). In this phase the $u_i$'s are decoupled ($\eta \dot{z}_i = 0$) and Eq. (2) becomes

$$\tau_u(t) = k(h_i - u_i) + k_0(V_0 t - u_i), \quad t > t_\eta. \qquad (6)$$

Implementing the velocity-strengthening friction (Eq. (4)), Eq. (6) admits an explicit solution[4,11]. More precisely, the time $t_R = \frac{A\sigma_N}{k_0 V_0}$ represents the long timescale associated with the afterslip of the layer U, and for $t_\eta < t < t_R$ one obtains

$$u_i(t) = u_i(t_0) + \rho_0 \log \left( 1 + D \frac{t - t_0}{t_R} \right), \qquad (7)$$

where $\rho_0 = \frac{A\sigma_N}{k+k_0}$ is a characteristic length and $D$ is a constant. Conversely, at later times $t > t_R$, the logarithmic motion becomes linear $u_i(t) \sim V_c t$ with $V_c = \frac{k_0}{k+k_0} V_0$.

To summarize, there are four timescales: (1) The slip timescale $t_s$, which characterizes the duration of a single earthquake, (2) $t_\eta$ related to the viscoelastic response in the layer U, (3) $t_R$ which corresponds to the posteismic phase, and (4) the inter-sequence timescale $t_d \sim \Delta h/V_c$ which corresponds to the typical waiting time between consecutive seismic sequences.

We assume an infinite time separation ($t_s \ll t_R \ll t_d$), which is a realistic approximation for geophysical parameters together with $t_s \ll t_\eta < t_R$. Under this hypothesis, three distinct phases are identified: coseismic phase ($t \sim t_s \ll t_\eta$), post-seismic phase $t \sim t_R$, ($t_s \ll t \ll t_d$) and interseismic phase $t \sim t_d \gg t_R$. Furthermore assuming that the local displacement $\Delta h$ is a constant independent of the position $i$, the temporal evolution of the model can be numerically implemented via a cellular automaton, for which each slip is infinitely fast. In this approximation, the dynamics of the layer $H$ at location $i$ is completely characterized by the two contributions to the stress acting on that site, namely the intra-layer stress $f_i(t) = k_h \nabla^2 h_i$ and the inter-layer stress $g_i = k(u_i - h_i)$. The sum $f_i + g_i$ is thus the total stress acting on block $i$. The details of the evolution of the variables $f_i$ and $g_i$ are given in the "Methods" section. In general, when $f_i + g_i \geq \tau_i^{th}$, there is a slip in the site $i$ and the stress evolves at $i$ and at nearest-neighboring sites $j$:

$$\begin{aligned} f_i(t) &\rightarrow f_i(t) - 4\Delta f \\ f_j(t) &\rightarrow f_j(t) + \Delta f \\ g_i(t) &\rightarrow g_i(t) - 4 k_h \Theta \Delta h \\ g_j(t) &\rightarrow g_j(t) + (\Theta - \epsilon)\Delta f \end{aligned} \qquad (8)$$

with $\Delta f = k_h \Delta h$, $\Theta = (1 - q_0)\frac{k}{4k_h}$ and $\epsilon = (1 - q_0 - 4q_1)\frac{k}{4k_h}$. The stress drop $\Delta f$ is extracted from a Gaussian distribution with average value $\langle \Delta f \rangle$ and standard deviation $\sigma$.

During the coseismic phase, the stress evolution is driven by all the slips in layer $H$. Conversely, during the post-seismic phase, the stress evolution is driven by the ductile behavior of the layer $U$. More precisely, since $u_i$ evolves according to Eq. (7), one has $g_i(t) = g_i(t_0)\Phi(t - t_0)$, where $\Phi(t)$ is a logarithmic decreasing function of time. During the interseismic phase, the stress $g_i(t)$ grows linearly in time at the very slow tectonic rate $k_0 V_c$.

Since the specific value of $\langle \Delta f \rangle$ is not relevant, we set $\langle \Delta f \rangle = 1$ and the model presents only three parameters: $\sigma$, $\Theta$, and $\epsilon$. The standard deviation $\sigma$ quantifies the level of friction heterogeneity, whereas $\Theta$ quantifies the elastic interaction between the two layers, and in the limiting case $\Theta = 0$ the layer H is decoupled from the layer U. Finally, the parameter $\epsilon \propto 1 - \sum_j q_{r_{ij}} = 1 - q_0 - 4q_1$ controls the amount of dissipation. In absence of friction in the layer $U$ ($\tau_u = 0$) and neglecting $k_0$ from Eq. (5), mechanical equilibrium imposes $\sum_j q_{r_{ij}} = 1$. However in general, for a finite $k_0$ and taking into account the inelastic deformations in the $U$ layer (the $z_i$ dynamics), $\sum_{j=1}^{N} q_{r_{ij}} < 1$. Accordingly, $\epsilon$ controls the value of an upper magnitude cutoff $m_U \simeq -1.5 \log_{10} \epsilon$ (see Supplementary Fig. 3). In the main text, we present results for a fixed value of $\epsilon = 0.008$ which allows us to explore a sufficiently large magnitude range without finite-size effects. The role of $\epsilon$ and of the system size $L$ is explicitly investigated in Supplementary Figures.

**Fundamental quantities and their statistical features in instrumental catalogs.** A key quantity is the seismic moment $M_0 = A\overline{D}$, where $A$ is the fractured area and $\overline{D}$ is the average displacement. In spring-block models, $A$ corresponds to the number of blocks which have slipped at least once during the earthquake and $M_0 = \sum_i n_i \Delta h$, where $n_i$ is the number of slips performed by the $i$th block during the earthquake. We next introduce the moment magnitude $m = (2/3)\log_{10} M_0$. In instrumental catalogs, $m$ is distributed according to the Gutenberg–Richter (GR) law: $P(m) \sim 10^{-bm}$, with quite a universal value[40] of $b \simeq 1$. It is worth noticing that the GR law corresponds to a power-law decay of the distribution of the

seismic moment $P(M_0) \sim M_0^{-1-2b/3}$. Furthermore, $M_0$ is related to the fractured area $A$ by the scaling relation $M_0 \sim A^{3/2}$ equivalent to the proportionality between $m$ and the logarithm of $A$, $m = \gamma_0 \log_{10} A + \text{cnst}$, with quite a universal coefficient $\gamma_0 = 1$[41,42].

**Comparison with previous spring-block models.** The description of a seismic fault in terms of spring and blocks was originally proposed by Burridge and Knopoff (BK)[33]. Bak and Tang[43] have enlightened the similarity between the BK model and the evolution of a simple cellular automaton model, the BTW model[44]. In the BTW model, the stress of each block increases in time with a constant rate $\dot{f}$, which models the tectonic loading, and when it reaches a uniform threshold $f\text{th}$, an earthquake starts by distributing stress to surrounding blocks. In the limit $\dot{f} \to 0$, once the bond network is assigned the BTW model does not have tunable parameters, and is usually considered the paradigmatic example of self-organized system, since it spontaneously evolves toward a state where the size of avalanches is power-law distributed. Identifying an avalanche with an earthquake, since the earthquake size is proportional to $M_0$, self-organized criticality provides a theoretical explanation for the GR law even, if it gives a too small, non-realistic value of $b$. Olami, Feder, and Christensen (OFC model)[45] have subsequently shown that, keeping the limit $\dot{f} \to 0$, the BK model can be exactly mapped in a cellular automaton. The model we present coincides with the OFC model in the limit cases $\Theta = 0$ and $\sigma = 0$ and, in turn, the OFC model coincides with the BTW model when $\epsilon = 0$. Interestingly, the OFC model presents an intermediate range of $\epsilon$ values such that $M_0$ is power-law distributed with a $b$ value close to one. On the other hand, for any finite value of $\epsilon$, in the OFC model $M_0 \propto A$ leading to $\gamma_0 = 2/3$ for the coefficient of the $m - \log A$ scaling, different from $\gamma_0 \simeq 1$ of instrumental catalogs.

Many modifications of the original OFC model have been proposed in the literature[2,46], and we group them into three classes: (i) those introducing a second timescale besides $\dot{f}$; (ii) those introducing heterogeneity in the friction thresholds $f\text{th}$; (iii) those introducing both a second timescale and friction heterogeneity. A second timescale is usually implemented in order to reproduce the temporal decay of the aftershock number which, indeed, can be attributed to a variety of time-dependent stress transfer mechanisms[47]. Major examples of class I models are those implementing a viscous relaxation[48–51] or a reductions in fault friction by means of RSF laws[50]. Concerning class II, the relevance of frictional heterogeneities in earthquake triggering has been deeply investigated[52] and, in particular, the OFC model with a random $f\text{th}$ corresponds to the quenched Edwards–Wilkinson (qEW) model[35,36,53]. This is a typical model for driven elastic interfaces in a random media and, in this case, it is well established that the seismic moment is power-law distributed with $b$ independently of the value of $\epsilon$[2]. Nevertheless, statistical patterns of seismic occurrence are better reproduced by class III models as shown in refs. [36,50,54–62].

According to the value of the parameters $\Theta$ and $\sigma$, our model can belong to the different classes. In particular, our conjecture is that class III models, and in particular the model we present with finite values of $\Theta > 0$ and $\sigma > 0$, belongs to the same universality class of seismic occurrence. This conjecture is supported by the results of the subsequent section. In particular, we observe that for finite values of $\Theta$ and $\sigma$ our model is very similar to the Viscoelastic quenched Edwards–Wilkinson (VqEW) model introduced by Jagla et al.[35]. The key difference lies in the functional form of $\Phi(t)$. Indeed in our model, the use of a realistic velocity-strengthening rheology induces a logarithmic variation of $\Phi(t)$ with time, which is the crucial ingredient leading to the Omori–Utsu hyperbolic decay of the aftershock rate. In the VqEW model instead, an exponential relaxation of $\Phi(t)$ is obtained.

**The magnitude distribution and the $m - \log A$ scaling.** For each earthquake we record the occurrence time $t$, the hypocentral coordinates $i$ (i.e., the coordinate of the block which nucleates the instability), the magnitude $m$, and the fractured area $A$. The simulated catalog contains about $10^7$ earthquakes; however, we exclude the first 10% of events so that results are independent of initial conditions. In the main text, we present results for different values of $\Theta$ and $\sigma$, keeping $\epsilon = 0.008$ fixed. The results for different $\epsilon$ are discussed in the Supplementary Notes.

The full separation of timescales allows us to clearly distinguish separate seismic sequences. We define a seismic sequence as the set of earthquakes triggered by the relaxation of the layer U, according to Eq. (7), i.e., the set of earthquakes triggered during the post-seismic phase. A new sequence starts at much later times when an earthquake is triggered during the interseismic phase with the slow stress rate increase $k_0 V_c$. Interestingly, as it is often observed in instrumental catalogs[63], this first earthquake in the sequence is not always the largest one. We adopt the convention used to classify events of real seismic sequences: the mainshock is the largest event in the sequence, the foreshocks are all events occurring before it and the aftershocks are all the subsequent ones. In Fig. 2a, we plot an excerpt of the whole catalog. The lag time between two consecutive sequences depends on the specific value of $t_d$.

Before studying the features of aftershocks and foreshocks, we investigate the behavior of the global catalog.

In Fig. 3, we plot the magnitude distribution $P(m)$ for different values of $\Theta$ and $\sigma$. In particular, the OFC model[45] (corresponding to $\Theta = \sigma = 0$) gives an exponential decay with $b = 0.12 \pm 0.02$ up to a system size-dependent upper cutoff $m_U$, whereas for the qEW model ($\Theta = 0$, $\sigma > 0$), we find $b = 0.40 \pm 0.02$ independently of $\epsilon$, with $m_U$ controlled by $\epsilon$. Surprisingly, even for small values of $\Theta$, the presence of the velocity-strengthening layer U induces a dramatic and robust change in the $b$ value. For $\Theta \geq 0.1$, in very good agreement with instrumental catalogs, we always observe $b = 1.06 \pm 0.05$ for intermediate magnitudes ranging from a lower cutoff $m_L$ related to lattice-specific details, up to an upper cutoff $m_U$. Keeping $\Theta > 0.1$ fixed we also find that the result $b = 1.06$ is independent of $\sigma$ (inset of Fig. 3), except for the singular choice $\sigma = 0$, where the magnitude distribution presents a non-monotonic behavior (not shown). The parameters $\Theta$ and $\sigma$ only affect the value of $m_U$, which increases with them (Fig. 3). In particular, $m_U$ tends to $m_L$ when $(\Theta, \sigma)$ are very small, shrinking to zero the range where $b \approx 1$. We also note an initial exponential decay $P(m) \sim 10^{-b'm}$ for small magnitudes (smaller than $m_L$), with $b'$ monotonically increasing with $\Theta$ from $b' = 0.65$ for $\Theta = 0.1$ to $b' = 1.42$, when $\Theta = 1$. In particular, we find an intermediate range of $\Theta$ values ($\Theta \in [0.4, 0.6]$), where $b' \simeq b$, i.e., for which the $b \simeq 1$ regime extends down to small magnitudes.

Realistic $b$ values of the GR law have already been found by Jagla et al.[58,59] in models of only one layer, but including viscoelastic couplings or aging effect in the static friction coefficients[36,54–62], which in both cases results in an effective additional degree of freedom per lattice site (all these models are spatially extended).

The coupling ($\Theta > 0$) with the layer U also allows us to recover the linear relation between $m$ and $\log(A)$. In the OFC model (and when $\epsilon \simeq 0$), a degree of freedom slips at most once by construction and therefore $M_0 \propto A$, $\gamma_0 = 2/3$ (Fig. 4). For the qEW model, the theory of depinning predicts $\gamma_0 = 2(1 + \zeta/d)/3$,

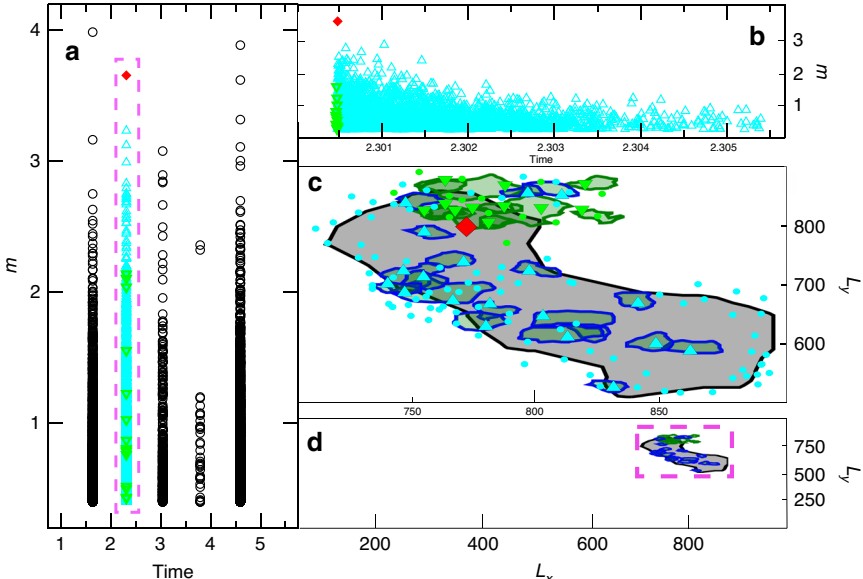

**Fig. 2 The numerical catalog. a** A typical example of a part of the simulated catalog containing five sequences. We plot the magnitude of each event $m$ versus its occurrence time in units of $t_d$. **b** A zoom on the second sequence plotted in panel (**a**). **c** We plot the contour of the area fractured by the mainshock (black line) of the $m > 1.5$ aftershocks (blue lines) and foreshocks (green lines) for the sequence plotted in panel (**b**). Red rhombus, cyan, and green triangles indicate the hypocentral location of the mainshock, of the $m < 1.5$ aftershocks and foreshocks, respectively. We include only aftershocks up to the time $t = 0.2t_R$ after the mainshock occurrence. **d** As in panel (**c**) for the whole fault plane, during the temporal window of the sequence considered in panel (**b**).

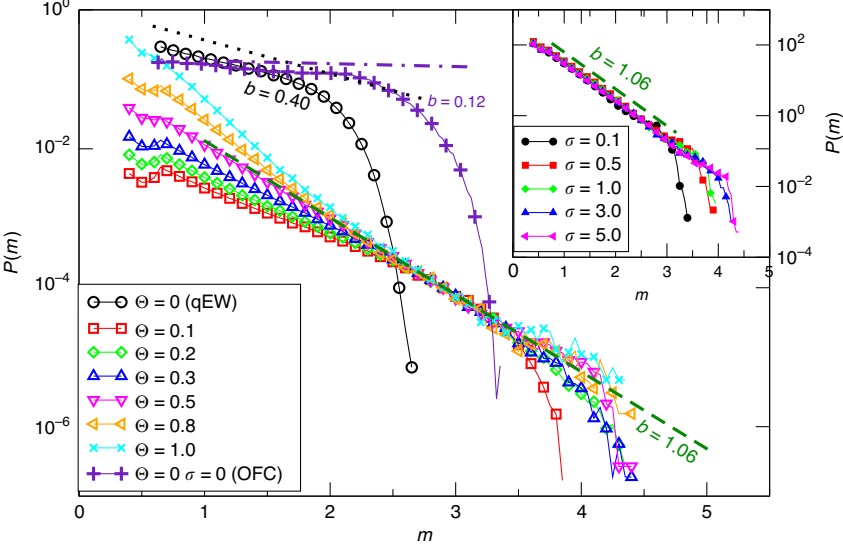

**Fig. 3 The magnitude distribution.** Magnitude distribution $P(m)$ for different values of $\Theta$ and $\sigma$. In the main panel, we fix $\sigma = 5.0$ and change $\Theta$, except for the OFC model with $\sigma = 0$. In the inset, we fix $\Theta = 0.5$ and change $\sigma$. Lines correspond to the GR law $P(m) \propto 10^{-bm}$ either with $b = 1.06$ (green dashed), consistent with the instrumental value, or $b = 0.12$ (turquoise dot-dashed) or $b = 0.40$ (black dotted).

with $d$ the dimension of the interface (here it coincides with the layer, $d = 2$) and $\zeta$ its roughness exponent. Here, we have $d = 2$ and $\zeta \sim 0.75$ (for long-range elasticity, $\zeta = 0$). This is consistent with the values measured: $\gamma_0 = 0.87 \pm 0.02$ (Fig. 4). When $\Theta > 0$ and $\sigma > 0$, we find a change to $\gamma_0 = 0.96 \pm 0.03$, independently of $\Theta$ and $\sigma$ (Fig. 4).

**Statistical features of aftershocks and foreshocks.** Let us now consider the properties of aftershocks and foreshocks. Results do not depend on the specific values of $\Theta$ and $\sigma$, thus we only present them for intermediate value of $\Theta = 0.5$ and for $\sigma = 5$. With these parameters, the GR law is obeyed over a sufficiently large magnitude range. The spatial organization of a typical fore-main-aftershock sequence is plotted in Fig. 2c, d, which presents the contour of the area fractured by a mainshock (here $m_M = 5.1$) and the contours of fracured area of the largest aftershocks and foreshocks ($m > 1.5$). Figure 2d just indicates that the whole sequence is concentrated in a narrow region of the fault plane close to the mainshock epicenter, while a zoom inside this region (Fig. 2c) provides details of the spatial organization of events. First of all we observe that most aftershocks occur close to the border of the mainshock's fractured area. This is consistent with the gap hypothesis according to which the increase of stress on the border of the fractured area triggers the aftershocks, whereas

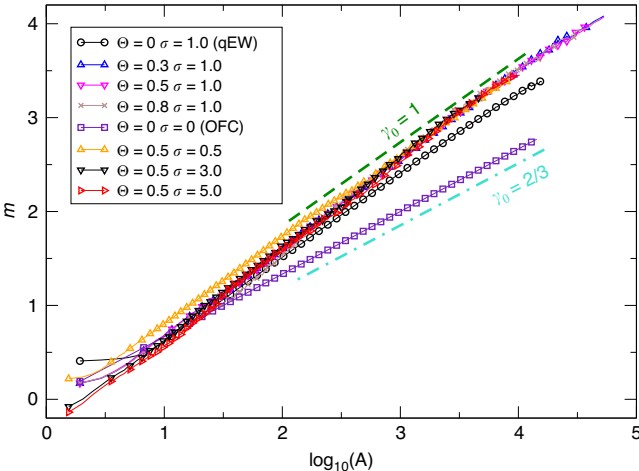

**Fig. 4 The $m$ – log$A$ scaling.** We plot the magnitude $m$ versus $log(A)$, for $\Theta = 0.5$, $\sigma = 5$, $L = 1000$ and $\epsilon = 0.008$. Lines correspond to the relation $m = \gamma_0 \log_{10}(A)$ with $\gamma_0 = 1$ (green dashed) and $\gamma_0 = 2/3$ (turquoise dot-dashed).

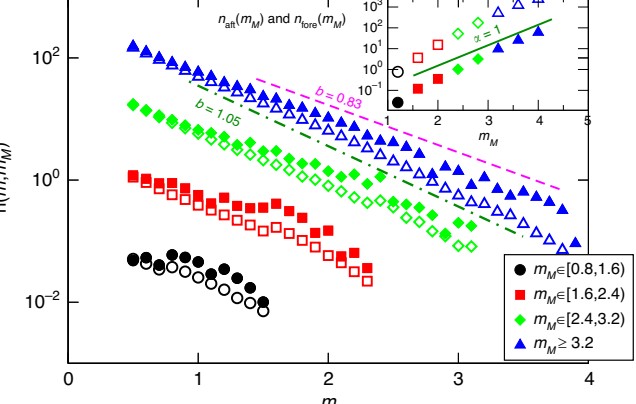

**Fig. 5 Aftershocks and foreshocks magnitude distributions.** We report the total number of aftershocks $n_{aft}(m, m_M)$ (open symbols) and foreshocks $n_{fore}(m, m_M)$ (filled symbols), with magnitude $m$, grouped by their mainshock's magnitude $m_M$ (see legend). We always consider $\Theta = 0.5$, $\sigma = 5$, $L = 1000$, and $\epsilon = 0.008$. Lines correspond to the GR law with $b = 1.05$ (green dot-dashed) and $b = 0.83$ (magenta dashed). (Inset) The aftershock number $n_{aft}(m_M)$ (empty symbols) and the foreshock number $n_{fore}(m_M)$ (filled symbols) versus $m_M$. The green line is the productivity law with $\alpha = 1$.

the stress reduction inside the fractured region strongly reduces their occurrence probability. This scenario is strongly supported by recent observations of the aftershock organization after big mainshocks[64]. The same analysis of Wetzler et al.[64] for the distribution of the aftershock hypocentral distance, from the contour of the mainshock fractured area, is presented in Supplementary Fig. 5. In our model also, foreshocks occur close to the border of the area that will be fractured by the mainshock. In order to be more quantitative, we plot (inset of Fig. 5) the number of aftershocks $n_{aft}(m_M)$ and foreshocks $n_{fore}(m_M)$ as a function of the mainshock magnitude. We find an exponential behavior $n_{aft}(m_M) \sim 10^{\alpha m_M}$, which is also observed in instrumental catalogs and known as the productivity law[65,66]. Also in this case we find quantitative agreement with the value $\alpha \simeq 1$ observed in instrumental catalogs. The inset of Fig. 5 also shows an exponential behavior $n_{fore}(m_M) \sim 10^{\alpha m_M}$ for the foreshock number with $\alpha \simeq 1$, a result also observed in instrumental catalogs[25,26]. We also find that the number of foreshocks is usually ~100 times smaller than the aftershock one, and we remark that only for the largest mainshock magnitude $m_M$, we do have a sufficient number of aftershocks ($n_{aft}(m_M) \gtrsim 1000$) to study their statistical features inside a single main-aftershock sequence. For this reason, to improve the statistics, we group sequences according to their mainshock's magnitude, as it is usually done in instrumental catalogs. More precisely, we consider the magnitude distribution of aftershocks (foreshocks) occurring after (before) a mainshock with magnitude $m \in (m_M, m_M + 1]$. Results (Fig. 5) confirm that aftershock magnitudes are distributed according to the GR law with $b \simeq 1$. Interestingly, we observe that also foreshocks follow the GR law but with a significantly smaller $b$ value $b \simeq 0.8$. This result is consistent with the existence of an inverse relation between b value and local stress level, as indicated by many laboratory measurements and field observations[67–69]. Accordingly, a smaller $b$ value (larger proportion of large earthquakes) is expected to be observed before the occurrence of the mainshock and close to its hypocenter, as a signature of high stress conditions. Indeed, several studies report the decrease of the $b$ value while approaching the mainshock, and identifies it as a precursory pattern which can improve mainshock forecasting[28–31]. Our study represents the first identification of this pattern in a mechanical model simultaneously presenting realistic features of aftershock occurrence. We further note that our measure of the $b$ value is neither biased by the foreshock selection criterion (since

we have a perfect separation of sequences) nor is it affected by problems of magnitude completeness (we have access to the smallest earthquakes), which are typical of instrumental catalogs and can be responsible for spurious behaviors of the $b$ value.

In Fig. 6, we plot the number of aftershocks (foreshocks) $n_{aft}(t|m_M)$ ($n_{fore}(t|m_M)$) as function of the time $t$ since (before) the mainshock with magnitude $m \in [m_M, m_M + 0.8]$, divided by the total number of mainshocks with $m \in [m_M, m_M + 0.8]$. We find that the aftershock organization in time follows the Omori–Utsu law $n_{aft}(t) \sim t^{-p}$ with $p = 1$ over several decades. The inverse Omori law[70,71] $n_{fore}(t) \sim t^{-p}$, with $p = 1$, is also found to characterize the temporal organization of foreshocks. It is worth noticing that, at variance with the aftershock occurrence, a clear temporal behavior cannot be extracted from a single foreshock sequence because of the very small number of foreshocks (we find at most 32 foreshocks during one sequence). Thus the inverse Omori law is only observed after stacking many sequences. The vertical shift of curves for different mainshock magnitudes is consistent with the productivity law, in agreement with the inset of Fig. 5. At short times, there is an abrupt transition from an about flat behavior to the $1/t$ decay. We expect that assuming a finite ratio $t_\eta/t_R$ would smooth this transition and help better reproduce instrumental observations.

In Fig. 7, we plot the density of aftershocks or foreshocks $\rho(\delta r, m_M)$ as a function of the distance $\delta r$ between their hypocenter and their mainshock's hypocenter, grouping events by intervals of mainshock magnitude $m_M \in [m_M, m_M + 0.8]$. There is a clear dependence on $m_M$, and at the same time for any $m_M$ the aftershocks and foreshocks share very similar spatial distributions, in agreement with instrumental findings[24,25,27]. Foreshocks occur mostly over the area fractured by the mainshock, supporting the idea that their spatial organization contains information on the size of the incoming mainshock (in that given region)[24,25]. Concretely, we find that $\rho(\delta r, m_M)$ obeys the scaling law $\rho(\delta r, m_M) = L(m_M)Q\left(\frac{\delta r}{L(m_M)}\right)$ with $L(M_m) \sim 10^{\gamma m_M}$ and $\gamma \simeq 0.57 \pm 0.05$. Similar collapses are observed in instrumental catalogs[24,25,72–74], although a smaller value $\gamma \simeq 0.5$ is usually observed. A second difference lies in the decay of the scaling function $Q(\delta r)$: in our model, it is exponential while power-law tails are reported in instrumental catalogs[74]. This overly fast decay can be attributed to our approximate modeling of elastic interactions within

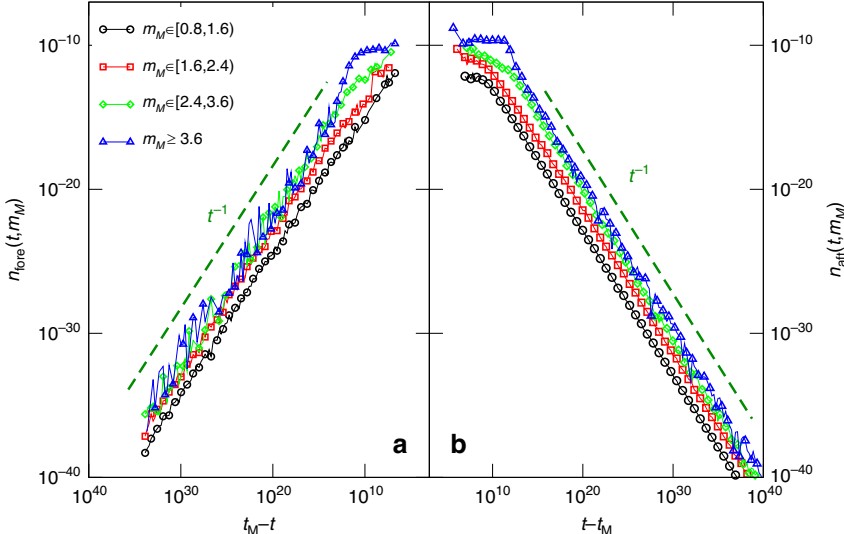

**Fig. 6 The direct and inverse Omori law.** The number of aftershocks $n_{aft}(t, m_M)$ (**b**) and the number of foreshocks $n_{fore}(t, m_M)$ (**a**) as function of the time $t$ since (and before) the mainshock occurrence. Different colors correspond to different mainshock magnitude classes $m_M$. The dashed line is the hyperbolic Omori–Utsu decay $1/t$. The wide range of the vertical scale makes difficult to appreciate the difference between the foreshock number and the corresponding aftershock number. This difference is better ennlightened by results plotted in the inset of Fig. 5. We always consider $\Theta = 0.5$, $\sigma = 5$, $L = 1000$, and $\epsilon = 0.008$.

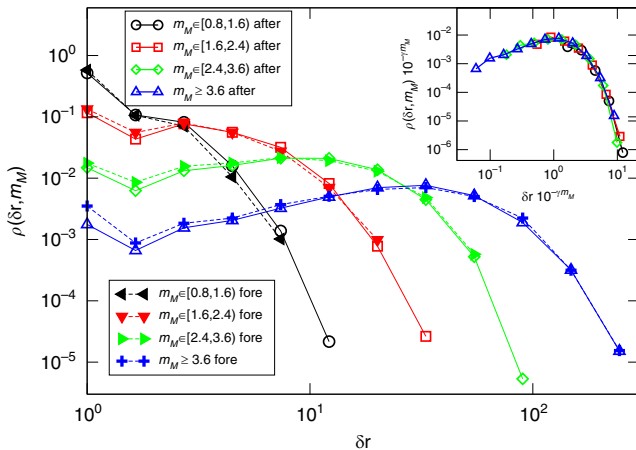

**Fig. 7 The spatial clustering of aftershocks and foreshocks.** The spatial density of aftershocks $\rho_{aft}(\delta r, m_M)$ (open symbols) and foreshocks $\rho_{fore}(\delta r, m_M)$ (filled symbols) as function of the hypocentral distance from the mainshock epicenter $\delta r$. Different colors correspond to different mainshock magnitude classes $m_M$. In the inset we show the same data after rescaling by the size of the aftershock area $L(m_M) = 10^{\gamma m_M}$ and $\gamma = 0.54$. We always consider $\Theta = 0.5$, $\sigma = 5$, $L = 1000$, and $\epsilon = 0.008$.

each layer, the correct long-range interaction expected from elasto-static theory[2] being replaced (see Eq. (1)) with the short-range term $k_h \nabla^2 h_i$. Indeed under this short-range approximation, aftershocks can be triggered only within or at the boundary of the rupture zone, as was shown in refs. [35,36]. Finally, we note that this spatial clustering law can be related to the m–logA scaling of Fig. 4, with $\gamma = 1/(2\gamma_0)$. Indeed since aftershocks are mostly distributed on the border of the area fractured by the mainshock, one has $L(m_M) \sim \sqrt{A} \sim \sqrt{10^{m_M/\gamma_0}} \sim 10^{m_M \gamma}$.

**Applications and improvements.** We have implemented a minimal model for earthquake triggering, modeling the interaction between the brittle part of the crust (an elastic and velocity-weakening region) and the ductile zone (a viscoelastic region with velocity-strengthening rheology). We assume short-range elasticity and infinite time separation, which allows to develop a cellular automaton model controlled by only two parameters, $\Theta$ and $\sigma$. Very interestingly, we find that as soon as $\Theta$ and $\sigma$ are sufficiently different from zero, we recover the established statistical features of aftershock occurrence, with realistic values of the parameters $b$, $\alpha$, $\gamma$, $p$. This robustness strongly suggests that our model captures the universality class of instrumental earthquakes. Our model thus provides useful insights on the mechanisms of aftershock triggering. For example, a deviation from the stationary behavior of the $b$ value is found during foreshock sequences, supporting its interpretation as a precursory pattern for the mainshock occurrence.

Although our model misses some features of instrumental earthquakes, such as the power-law decay of the spatial density $\rho$, it can be very useful. Thanks to its simplicity, we can easily produce very complete synthetic catalogs to test forecasting hypothesis, or mechanisms of stress evolution and how it is related to foreshocks, mainshocks, or aftershocks. It is also possible to extend the single-fault model presented in this study to a more realistic description as a network of faults. One could then study the interaction between different branches of the network.

## Methods
**Derivation of the cellular automaton rules.** We consider two square layers of sides $L = 1000$. In our lattice geometry, there are exactly four neighbors $j$ for each site $i$: the stress diffusion terms of the type $(\nabla^2 h)_i$ at site $i$ with positions $x, y$ thus stand for $(\nabla^2 h)_{x,y} = (h_{x+1,y} + h_{x-1,y} + h_{x,y+1} + h_{x,y-1} - 4h_{x,y})$. We use absorbing boundary conditions ($h = 0$ is fixed at the boundary), which means that some stress is absorbed at the boundaries, and the slip cannot propagate further.

We now recall the main assumptions of our continuous model, before explicitly deriving the corresponding cellular automaton. These assumptions are summarized in the mechanical sketch of Fig. 1, from which the equations can be derived. The stress at site $i$ in the layer $H$ is the sum of intra-layer and inter-layer stresses, respectively:

$$f_i = k_h \nabla^2 h_i \qquad (9)$$

$$g_i = k(u_i - h_i). \qquad (10)$$

The total stress $f_i + g_i$ at site $i$ is balanced by a velocity-weakening (Coulomb failure style) friction force $\tau_h$, which takes a new random value, denoted $\tau_i^{th}$, after each slip:

$$\tau_h = \tau_i^{th} \sim G(\tau) \sim \mathcal{N}(1, \sigma) \qquad (11)$$

where $G(\tau)$ is a gaussian distribution with average 1 and standard deviation $\sigma$. As long as $\tau_h \geq f_i + g_i$ the site $i$ is pinned, that is $\dot{h}_i = 0$. The constitutive equations operate on various timescales:

$$\tau_h \geq f_i + g_i \qquad \text{slip timescale } t_s \qquad (12)$$

$$\tau_{u_i} = k_u(\nabla^2 u_i - z_i) + k(h_i - u_i) + k_0(V_0 t - u_i) \qquad \text{slip timescale } t_s \qquad (13)$$

$$\eta \dot{z}_i = k_u(\nabla^2 u_i - z_i), \qquad \text{visco-elastic timescale } t_\eta = \frac{\eta}{k_u} \qquad (14)$$

$$\tau_{u_i}(t) = \sigma_N\left(\mu_c + A \log \frac{\dot{u}_i(t)}{V_c}\right), \qquad \text{relaxation timescale } t_R = \frac{\rho_0}{V_c} = \frac{A\sigma_N}{k_0 V_0} \qquad (15)$$

The first two equations are the force balance between applied stresses and local friction force, and are thus instantaneous. The third is the internal stress dynamics of the viscoelastic layer $U$, evolving over an intermediate timescale $t_\eta$. The fourth is the time evolution of the velocity-strengthening friction, slowly evolving over a timescale $t_R$. We recall the constants: $V_c = \frac{k_0}{k+k_0} V_0$, $\rho_0 = \frac{A\sigma_N}{k+k_0}$.

- Initialization: At time $t = 0$, we assign a local frictional threshold $\tau^{\text{th}}$ extracted from $G(\tau)$. We also choose the initial value $f_i(0)$ of the local stress at random in the interval $(0, \tau_i^{\text{th}})$ and suppose that at $u_i(0) = h_i(0)$ in all sites.

- Interseismic phase: At time scales larger than $t_s$, $t_\eta$, $t_R$, we have $\dot{u}_i = V_c$ and the equations above simplify. Using Eq. (15), we get $\tau_u = \mu_c$. At these long timescales ($t \gg t_\eta$), we have $\dot{z}_i = 0$ so that using Eq. (14), $z_i = \nabla^2 u_i$. Using Eq. (13), this combines to yield $k_0 V_0 t = (k + k_0)V_c t$, which explains the necessary definition $V_c = \frac{k_0 V_0}{k+k_0}$. We finally have $f_i + g_i = \text{const.} + kV_c t$, and using Eq. (12), we can compute the distance to failure (time before failure):

$$t_i^{\text{(drive)}} = \frac{\tau_i^{\text{th}} - f_i(t_0) - g_i(t_0)}{kV_c} \qquad (16)$$

with $t_0$ the time at the beginning of this phase. The site $i^*$ corresponding to the smallest value of $t_i^{\text{(drive)}}$ is thus identified as the hypocenter of the next earthquake. An amount of stress $\tau_{i^*}^{\text{th}} - f_{i^*}(t_0) - g_{i^*}(t_0)$ is then added to all sites and the coseismic phase is entered, with exactly one site being unstable (the one where $f_{i^*}(t) + g_{i^*}(t) = \tau_{i^*}^{\text{th}}$).

- Coseismic phase: Each site with $f_i(t) + g_i(t) \geq \tau_i^{\text{th}}$ is unstable and slips of a constant amount $\Delta h$, $h_i \to h_i + \Delta h$. A slip in the layer $H$ at site $i$ induces a coseismic slip $u_j \to u_j + q_{r_{ij}}\Delta h$ inside the $U$ layer. As explained in the main text, we set $q_r = 0$ for $r > 1$, i.e., we only keep the local coseismic slip $q_{r_{ii}} = q_0 > 0$ and the nearest-neighbor coseismic slip $q_{r_{ij}} = q_1 > 0$ (when $|r_{ij}| = 1$). Because of the ductile nature of the layer $U$ there is some dissipation occurring during the coseismic slip, in the sense that the total coseismic slip is less than the slip: $\bar{\epsilon} = 1 - q_0 - 4q_1 > 0$ ($\bar{\epsilon} = 0$ would be the dissipationless case). This coseismic slip is considered instantaneous and corresponds to the following stress evolution, for the site $i$ itself and for its first neighbors $j$:

$$f_i(t) \to f_i(t) - 4k_h \Delta h \qquad (17)$$

$$f_j(t) \to f_j(t) + k_h \Delta h \qquad (18)$$

$$g_i(t) \to g_i(t) + k(q_0 - 1)\Delta h \qquad (19)$$

$$g_j(t) \to g_j(t) + kq_1 \Delta h \qquad (20)$$

At this timescale, the internal degrees of freedom $z_i$ are fixed and do not evolve. By introducing the parameters

$$\Theta = (1 - q_0)\frac{k}{4k_h}, \qquad (21)$$

$$\epsilon = (1 - q_0 - 4q_1)\frac{k}{4k_h} = \bar{\epsilon}\frac{k}{4k_h}, \qquad (22)$$

we can factorize:

$$f_i(t) \to f_i(t) - 4k_h \Delta h \qquad (23)$$

$$f_j(t) \to f_j(t) + k_h \Delta h \qquad (24)$$

$$g_i(t) \to g_i(t) - 4k_h \Theta \Delta h \qquad (25)$$

$$g_j(t) \to g_j(t) + (\Theta - \epsilon)k_h \Delta h \qquad (26)$$

Which shows that the coseismic slip stabilizes $g_i$ but increases the $g_j$ stresses. After a slip, the block $h_i$ is in a different frictional condition, i.e., a new value of $\tau_i^{\text{th}}$ is extracted from the distribution $G(\tau)$. If $\tau_i^{\text{th}}$ is such that $f_i(t) + g_i(t) \geq \tau_i^{\text{th}}$ then the process of Eqs. (23)–(26) is iterated immediately, until $f_i(t) + g_i(t) < \tau_i^{\text{th}}$. Because of the stress redistribution, nearest-neighbor sites $j$ can be unstable and slip at the

same time. In practice, we perform the updates of Eqs. (23)–(26) on all sites for which $f_j(t) + g_j(t) \geq \tau_j^{\text{th}}$, until all sites satisfy $f_j(t) + g_j(t) < \tau_j^{\text{th}}$.

We follow a sequential updating scheme, which implies the slip of just one unstable block at a time. Preliminary results with an updating scheme where all unstable blocks simultaneously slip indicate no important difference.

Shortly after the end of the earthquake, the viscoelastic couplings (internal degrees of freedom $z_i$ of the layer $U$) relax their stress (over a timescale $t_\eta = \eta/k_u$), which in practice means that $\dot{z} = 0 = k_u(\nabla^2 u_i - z_i)$. As we explained in the main text, the way in which this relaxation affects the stress in the layer $U$ has already been included implicitly in the coslip dynamics, via the coefficients $q_0$, $q_1$, so that on longer timescales we can simply consider that $k_u(\nabla^2 u_i - z_i) = 0$. In particular, Eq. (5) simplifies into Eq. (6), i.e., the blocks $u_i$ become independent.

- Post-seismic phase: At the end of an avalanche, the $h_i$ are stuck, so that the intra-layer stress $f_i$ remains constant. However, the $g_i$ may evolve. Indeed, the $u_i$ are subject to a velocity-strengthening rheology and evolve according to Eq. (6), which can be integrated to give Eq. (7), that we recall here:

$u_i(t) = u_i(t_0) + \rho_0 \log\left(1 + D\frac{t-t_0}{t_R}\right)$, where $D = \exp(-g(t_0)/A\sigma_N)$ is a constant (over time, but different for each site). This solution can be checked, using Eq. (15) on one side and Eq. (6) on the other. One may also consult our solution for the case of a two-blocks model[11], which is the same since all sites $u_i$ evolve independently during the afterslip (in the two-block model there was only one block $u$). Writing $g_i(t) = g_i(t_0) + k(u_i(t) - u_i(t_0))$, we can identify

$$g_i(t) = g_i(t_0)\Phi(t - t_0) \qquad (27)$$

with

$$\Phi(t - t_0) = 1 - \frac{k}{k + k_0}\frac{\log\left(1 + D\frac{t-t_0}{t_R}\right)}{\log(D)}, \qquad (28)$$

and the local stress value evolves, at times $t \geq t_0$, according to the equation:

$$f_i(t) + g_i(t) = f_i(t_0) + g_i(t_0)(1 - \Phi(t - t_0)). \qquad (29)$$

We note that this happens independently in all the sites (there is no stress transfer between sites, which makes the evolution very simple to compute). It is important to remark that $\Phi(t)$ is a monotonically decreasing function of $t$. For $g_i(t_0) < 0$ (which does happen), $D$ is large and at times $t - t_0 \gg t_R$, $\Phi(t - t_0) \sim 0$. Thus $\Phi$ decreases from 1 to $\approx 0$. This relaxation of the inter-layer stress $g_i(t)$ happens to all sites simultaneously. If for a site $i$, $f_i(t_0) > \tau_i^{\text{th}}$, there will be a time $t_{\text{aft}} > t_0$ such that $f_i(t_{\text{aft}}) + g_i(t_{\text{aft}}) = \tau_i^{\text{th}}$. For some sites (depending on the slips dynamics), this condition is not fulfilled and no such time exists.

Computationally, we compute the time $t_{\text{aft}}$ for all sites where it is defined by inverting Eq. (29). Then we pick the smallest $t_{\text{aft}}$, that we may call $t_{\text{aft}}^*$, and relax the stress in all sites according to Eq. (29) using $t = t_{\text{aft}}^*$. At this point, there is thus exactly one site that became unstable (the one with the smallest $t_{\text{aft}}$), i.e., a new earthquake is triggered. We then proceed with the coseismic phase, until the earthquake is complete. This is the physical mechanism which triggers aftershocks.

After a number of aftershocks, there is a point where no site fulfills the condition $f_i(t_0) > \tau_i^{\text{th}}$, i.e., no time $t_{\text{aft}}$ can be defined. In that case, we perform relaxation "for infinite time" (several $O(t_R)$), or more concretely we set $g_i = 0$ at all sites. At this point, the fore-main-aftershock sequence is finished and the interseismic phase resumes, triggering a new sequence.

## Data availability
Numerical data that support the findings of this study are available from the corresponding author upon reasonable request.

## Code availability
The source code of the numerical model is available from the corresponding author.

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

## Acknowledgements

We thanks Eduardo Jagla for useful discussions. This research activity has been supported by the Program VAnviteLli pEr la RicErca: VALERE 2019, financed by the University of Campania "L. Vanvitelli". E.L. and G.P. also acknowledge support from MIUR-PRIN project "Coarse-grained description for non-equilibrium systems and transport phenomena (CO-NEST)" no. 201798CZL.

## Author contributions

G.P., E.L., F.L., and A.R. have all contributed extensively to numerical simulations, data analysis, and to write the paper.

## Competing interests

The authors declare no competing interest.
