## [Peer Review File · Nature Communications]

Reviewers' Comments:

Reviewer #1:

Remarks to the Author:

Nature Communications manuscript NCOMMS-20-01427, "The influence of the brittle-ductile transition zone on aftershock and foreshock occurrence" by G. Petrillo et al.

The manuscript builds on a 2 layer cellular automaton to mimics the changes of rheological behavior of the crust with increasing depth, from velocity weakening to velocity strengthening. The model succeeds in reproducing the observed statistical features of both aftershocks and foreshocks. The most impressive outputs lie in the b value change in foreshocks as compared to aftershocks distribution, respectively. This latter result, which reinforces the intensely debated robustness of possible observed foreshocks is missing to be explicitly described in the current abstract version. When the overall manuscript is of high quality, i suggest reshaping parts of the manuscript will enhance its impacts. This task is an low-risk high gain recommendation that should be quickly fulfilled by the author group skills.

First, the mapping of the model hypothesis on the current seismicity challenges and context is not clear enough. As example, when pointing on the key role of afterslip in earthquake triggering, the authors appear to bound the seismicity in the 5-15 km range. Such a depth statement is only true for the "continental" earthquakes, it does not apply for subduction zone seismicity. However, several cases discussed in the text and the referenced literature focus on subduction zone quakes. Accordingly, the authors should clarify the database they use all along the manuscript to calibrate the model outputs.

Second, the comparison with previous spring-blocks model should (i) be accessible to a broader audience and (ii) comprehensively review the current knowledge. As an example, Pelletier (2000) and Jagla et al. model both reproduce the submitted manuscript results closely. More focused text on which control parameters support advances relative to previous simulations should be included in a revised version. More technically, it is known (e.g., the review by Arcangelis et al. 2016) that some models fail to reproduced observations for the whole scale of earthquake size. Why are the simulation results bounded to $M_{max}=4$ should be discussed (a few lines and figure(s) in supplementary materials are welcome)? Which are the largest/smallest possible event in the 10000×100000 lattice size the authors use should be discussed?

Third, from the result section, figure 3-5 demonstrates there is a limited range of theta and sigma parameter values that reproduces the observed scaling laws for aftershocks and foreshocks. It contradicts the abstract result described as "recovered at a quantitative level without any fine-tuning."

specific comments:

p3, All the sentences on foreshocks are specific to rare cases. The text should be tampered down accordingly.

p3, two (over 3) reference for before-shock-slow-slip apply for subduction cases. How does the model works for these cases is not clear when authors describe the universality of aftershocks and foreshocks patterns for shallows seismicity solely?

p3, the reference to Gulia and Wiemer is very controversial and nonuniversal at all. First, the authors suggest that the observed pattern is deterministic rather than statistical. Second, only two cases are analysed by Gulia and Wiemer, with different space-time-magnitude window for each event, without justification (e.g. Brodsky, E. E. (2019). Determining whether the worst earthquake has passed)

p9 top, "have earthquakes as large as the whole system, something that happens in observational data."

The authors must specify what the "system" that is supposed to correspond to observational data

is. Because earthquake size remains non-predictable, observational data are related to a given seismic network for which no system can be isolated from each other. Furthermore, is it not the "coupling at different scales" the basis of the author's primary hypothesis ("The organization in time, space and energy of aftershocks is characterized by scaling behaviors with exponents which are quite universal"). In such a context, "earthquake as large as the system size" must be clarified. Note here that on the result section, the M_{max} is bounded to $M=4$ without information on the simulated system size?

p9 comparison with previous spring-blocks model:

- magnitude definition and M_{max} $M_0=f(\text{slip})$ versus $M=f(\text{Sliding block})$?

The model of Pelletier 2000 presents seismic activity consistent with the GR law for small events whereas the large magnitude behavior changes from supercritical behavior, for large seismic couplings, to a GR distribution with a magnitude cutoff, for low seismic couplings. Similarly (De archangels et al. 2016 points on "BK models implementing state-rate friction laws have also been widely studied. Results show that the magnitude distribution is strongly affected by the friction parameters. Under the condition of so-called 'strong friction instability' a GR behavior is found at small magnitudes with $b \approx 1$, followed by an almost flat behavior reminiscent of the peak structure found in the super-critical regime. A double peak distribution is conversely observed in the opposite condition of 'weak friction instability' These references question the ability of the author model to reproduce the full range of magnitude. Why are distribution results bounded to $M1-4$ range ? Which are the largest/smallest possible event in the 10000×100000 lattice the authors use?

Results

- Figure5-7: there is an indication for neither θ nor σ values that are used for these simulations?

-Figure 6: the number of fore-aftershocks should not be the same, as pointed in the text and on (inset) figure 5?

-p13 aftershock location:

"First of all, we observe that most aftershocks occur close to the border of the mainshock's fractured area.... . This scenario is strongly supported by recent observations of the aftershock organization after big mainshocks 59."

Most of the #59 reference earthquakes are subduction zone earthquakes below 15 km depth (see p2). Are the patterns of "universality" classes still stable with depth, is not discussed? Some recent papers point the others way (e.g., Hainzl et al. JGR 2019)

-A modified figure 2 to present the results the same way as cited in #59 reference is more than welcome?

Reviewer #2:

Remarks to the Author:

This work, based on a novel geophysical numerical model, shows that a combination of two rheological behaviors, velocity weakening and velocity strengthening, is a mechanism generating the statistical features of aftershocks and foreshocks. These features include a smaller b -value for foreshocks and aftershock regions scaling with main shock magnitude. To my knowledge, these results are original. Reproducing complex statistical spatio-temporal features of seismicity with a model devised from physical laws and quite simplified (to a justified level) is remarkable.

The text is well written (see however points below) and scientifically sound. The results are very interesting and the claim of this work is important. In my opinion there are the ingredients for publishing in Nat. Comm., provided that the following points are addressed convincingly by the Authors.

1. By the notation in text (one-dimensional indices "i" and "j"), and by looking at figure 1, the impression is that one deals with a $2 \times N$ model. The sentence ending with "organized on a

rectangular lattice (see Fig. 1)" seems to confirm this interpretation. However, figure 2 shows a square lattice with sides L_x and L_y not introduced in the main text. Only in the Methods this becomes clear. The text should be improved to better describe the geometry of the model.

Having settled that the model is composed by two 2d carpets on top of each other, it becomes less clear how this is related to the depth in the crust. The Authors argue that the changing with depth of rheological properties of the crust is the feature that justifies the introduction of both velocity weakening and velocity strengthening. How is this represented in the model? Is it a model of a single fault or of a tectonic region?

The excellent performance of the model invites to think of some alternative interpretation: Is it possible that weakening and velocity strengthening are both taking place at the two sides of a fault (in different measure) at every depth?

2. The "without any fine-tuning" claim in the abstract should be better explained. In the text it is not obvious if the third parameter epsilon (for dissipation) is exactly set to zero or is kept at an arbitrarily small value. The readers should be able to find all information for reproducing the results.

At page 12 " Results do not depend on the specific values of Θ and σ , thus we only present them for intermediate values: $\Theta = 0.5$, $\sigma = 5$." should be better justified.

The range of variability for the disorder is chosen by the authors and $\sigma=5$ seems rather at its top than at intermediate range.

$\Theta=0.5$ gives the better Gutenberg-Richter law in figure 3 and indeed it becomes the value over which the Authors focus in other figures. I do not understand the mismatch in cutoff between the curves for $\Theta=0.5$, $\sigma=5$ in the inset (orange data) of figure 3 and in its main panel (magenta). They should be the same curve, I believe.

3. At page 9 there is a hint at the BTW model. This seems a bit limited. The introduction should include a broader perspective giving the proper credit to the mechanism of self-organized criticality as the origin of avalanches with power laws.

4. In the Methods, it is not clear if the model is updated sequentially or in parallel. There would be no difference if the model were Abelian as the BTW model, which is probably requiring a hard mathematical proof. The statement "The precise synchronization of the updates does not matter" makes sense only if the model is Abelian.

Minor points:

* The caption of figure 4 does not comment on its inset.

* Sometimes Θ becomes θ in the text and figures.

* Caption 1: " each fault modeled as" → " each fault is modeled as"

* Page 13: "largest mainshock magnitude m_M do we have a sufficient number" → we do have

* Page 21: " Indeed, the ui are subject to a velocity-weakening rheology", should be strengthening

Marco Baiesi,
Dipartimento di Fisica e Astronomia "Galileo Galilei",
Universita' di Padova,
Italy

Answer to Reviewer #1
#####

We thank the reviewer for the positive evaluation of our manuscript and for very appropriate comments and remarks. We have implemented all of them in the revised version of the manuscript. In particular we thank the reviewer for considering the change in foreshocks as compared to aftershocks distribution an ``impressive output''. Following the referee's suggestion we explicitly discuss this result in the revised abstract.

In the following we answer to the reviewer's general comments

1) The reviewer writes:

``First, the mapping of the model hypothesis on the current seismicity challenges and context is not clear enough. As example, when pointing on the key role of afterslip in earthquake triggering, the authors appear to bound the seismicity in the 5-15 km range. Such a depth statement is only true for the "continental" earthquakes, it does not apply for subduction zone seismicity. However, several cases discussed in the text and the referenced literature focus on subduction zone quakes. Accordingly, the authors should clarify the database they use all along the manuscript to calibrate the model outputs.''

Our model assumes that the seismic fault is a velocity-weakening layer embedded in a velocity-strengthening medium, where afterslip relaxation takes place. We describe the velocity-strengthening medium as a second layer elastically coupled to the fault one. The relative position of the two layers is not relevant and therefore our description applies to any tectonic context and in particular to subduction zones. In particular it captures the mechanism for aftershock migration proposed for the 2015 Mw8.3 Illapel earthquake in Central Chile [10] and for the Tohoku earthquake [11]. We agree with the referee that this was not clear from our introductory paragraph where we described only continental seismicity. In the revised version we have modified this paragraph and clarified that our description applies to different tectonic contexts.

2) The reviewer writes:

``Second, the comparison with previous spring-blocks model should (i) be accessible to a broader audience and (ii) comprehensively review the current knowledge. As an example, Pelletier (2000) and Jagla et al. model both reproduce the submitted manuscript results closely. More focused text on which control parameters support advances relative to previous simulations should be included in a revised version.''

We regret that our comparison with previous spring-block models was not clear, in particular we missed to refer to the interesting manuscript by Pelletier (new ref.[51]). In the new version of the manuscript we improved the Section ``Comparison with previous spring-block models': The spring-block models are grouped in three different classes according to the

presence/absence of heterogeneity and stress relaxation. Our conjecture is that each group behaves as a sort of universality class in the sense that many statistical features are common to all models belonging to the same group. Our model presents both friction heterogeneity and stress relaxation as the ones proposed by Pelletier and by Jagla et al. So it belongs to the group (iii), that we show to reproduce the features of real seismicity. Note that previous works reproduce only a small number of the seismic features: the productivity law, the m -log A scaling and the scaling of spatial clustering of aftershocks and foreshocks have been investigated for the first time in the present study. These features request large system sizes and statistics which are accessible with our very efficient cellular automata algorithm. Numerical simulations by Pelletier, for instance, have investigated only small size systems of 64×64 . Furthermore, our algorithm allows us to explore the influence of the different ingredients which are controlled by the two model parameters θ and σ and changing these values we can explore the three different universality classes.

2a) The reviewer writes:

``More technically, it is known (e.g., the review by Arcangelis et al. 2016) that some models fail to reproduce observations for the whole scale of earthquake size. Why are the simulation results bounded to $M_{\max}=4$ should be discussed (a few lines and figure(s) in supplementary materials are welcome)? Which are the largest/smallest possible event in the 10000×10000 lattice size the authors use should be discussed?''

The system size fixes an upper bound on the maximum observed magnitude M_{\max} but, for our results, M_{\max} is mostly controlled by the parameter ϵ , which corresponds to the amount of dissipation in the system. In order to avoid finite size effects we have presented numerical simulations for a finite value of $\epsilon=0.008$. For this choice of ϵ , indeed, the maximum recorded avalanches span at most over a region of order 200×200 which is significantly smaller than the considered system size 1000×1000 . The absence of finite size effects is illustrated in the new Suppl.Fig.1. It is quite evident that, for our choice of ϵ the magnitude distribution is only weakly affected by the system size L . At the same time, in the new Suppl.Fig.2 we present the magnitude distribution for different values of ϵ and we discuss as the upper cut-off magnitude $m_{\{U\}}$ scales with ϵ (Supp. Fig.3). We wish to stress that a finite value of ϵ is a realistic choice since, during an earthquake, a relevant percentage of the accumulated energy is dissipated in heat. Furthermore, the CPU time requested for the evolution of an earthquake of magnitude m is roughly proportional to $10^{\{3/2 m\}}$, therefore a smaller value of m_{\max} allows us to achieve a huge statistics. At the same time, the system size 1000×1000 is sufficiently large to study the spatial distribution of aftershocks and the spatial correlation among different earthquakes, without relevant finite size effects. In all cases, except for the cut-off on the maximum magnitude, we find that also other findings appear quite independent of ϵ , as shown in the new supplementary materials (Supp. Fig.4). We also want to point to the referee attention that, also in the limit case $\epsilon=0$, when one can have avalanches involving the whole system, it is difficult to find the precise value of M_{\max} for a fixed L . Indeed,

for an avalanche involving the whole system we have that the seismic moment is given by $M_0 = L^2 \times \overline{D}$ where \overline{D} is the average displacement leading to $M_{max} = (4/3) \log_{10}(L) + (2/3) \log_{10}(D)$, but the precise value of \overline{D} is difficult to establish.

In the supplementary materials (Supp. Figs. 1-4) of the revised version we explicitly discuss the role of ϵ and of L on the maximum magnitude and on the other results presented in the main manuscript.

3) The reviewer writes:

``Third, from the result section, figure 3-5 demonstrates there is a limited range of theta and sigma parameter values that reproduces the observed scaling laws for aftershocks and foreshocks. It contradicts the abstract result described as "recovered at a quantitative level without any fine-tuning."''

Data of Fig.3 shows that for all values of θ and for $\sigma \geq 1$, we always observe a magnitude range where data follows the GR law with a b -value ≈ 1 . Our interpretation, is that small or large values of θ introduce long crossovers which hide the true asymptotic value $b \approx 1$. In other words, our conjecture is that if we were able to simulate large system with very small ϵ values, after a transient regime at small magnitudes, we expect that a clear regime with $b \approx 1$ would be observed also for $\theta = 0.1$ or $\theta = 1$. However, we agree with the reviewer that presented data do not strongly support our interpretation and we have modified the sentence in the abstract of the revised manuscript.

Answer to specific comments:

``p3, All the sentences on foreshocks are specific to rare cases. The text should be tampered down accordingly.''

We agree with the reviewer and we have modified the text accordingly.

``p3, two (over 3) reference for before-shock-slow-slip apply for subduction cases. How does the model works for these cases is not clear when authors describe the universality of aftershocks and foreshocks patterns for shallows seismicity solely?''

We have addressed this remark in our answer at point 1.

``p3, the reference to Gulia and Wiemer is very controversial and nonuniversal at all. First, the authors suggest that the observed pattern is deterministic rather than statistical. Second, only two cases are analysed by Gulia and Wiemer, with different space-time-magnitude window for each event, without justification (e.g. Brodsky, E. E. (2019). Determining whether the worst earthquake has passed)''

We briefly discuss the limit of the results by Gulia and Wiemer in the revised version and we quote the comment by Brodsky.

``p9 top, "have earthquakes as large as the whole system, something that happens in observational data."

The authors must specify what the "system" that is supposed to correspond

to observational data is. Because earthquake size remains non-predictable, observational data are related to a given seismic network for which no system can be isolated from each other. Furthermore, is it not the "coupling at different scales" the basis of the author's primary hypothesis ("The organization in time, space and energy of aftershocks is characterized by scaling behaviors with exponents which are quite universal"). In such a context, "earthquake as large as the system size" must be clarified. Note here that on the result section, the Mmax is bounded to M=4 without information on the simulated system size?'' We totally agree with the referee on this point and the sentence "earthquake as large as the system size" is completely misleading. For this reason we have removed it in the revised version of the manuscript and at the same time we explicitly discuss the role of ϵ and of the system size in the Supplementary Materials.

``p9 comparison with previous spring-blocks model: - magnitude definition and Mmax Mo=f(slip) versus M=f(Sliding block)?
The model of Pelletier 2000 presents seismic activity consistent with the GR law for small events whereas the large magnitude behavior changes from supercritical behavior, for large seismic couplings, to a GR distribution with a magnitude cutoff, for low seismic couplings. Similarly (De archangels et al. 2016 points on "BK models implementing stateâ€"rate friction laws have also been widely studied. Results show that the magnitude distribution is strongly affected by the friction parameters. Under the condition of so-called 'strong friction instability' a GR behavior is found at small magnitudes with $b \approx 1$, followed by an almost flat behavior reminiscent of the peak structure found in the super-critical regime. A double peak distribution is conversely observed in the opposite condition of 'weak friction instability' These references question the ability of the author model to reproduce the full range of magnitude. Why are distribution results bounded to M1-4 range ? Which are the largest/smallest possible event in the 10000x100000 lattice the authors use?''

We have addressed this remark in our answer at point 2. We wish to remark that in our simulations we always take a finite ϵ which accounts for a finite degree of dissipation in the system. In our cases we never observe super-critical behavior but we always find that the magnitude-distribution is tempered after a given upper magnitude M_U which depends on ϵ . For very small ϵ (see Suppl. Fig.3), when we find earthquakes involving the whole system $A \sim L^2$, we also find a peak in the magnitude distribution as in super-critical behavior. Nevertheless we mostly focus on the regime where finite size effects are not present. In the revised version we give a more precise definition of the magnitude and in the supplementary materials we discuss how parameters affect the upper magnitude cut-off.

----- Results -----

``- Figure5-7: there is an indication for neither theta nor sigma values that are used for these simulations?''

Thanks for the remark. In the revised version we now specify that results are for $\theta=0.5$ and $\sigma=5$ in the figure caption.

``-Figure 6: the number of fore-aftershocks should not be the same, as pointed in the text and on (inset) figure 5?''

The reviewer is right, the foreshock number is always smaller the

aftershock one, as correctly indicated in the inset of Fig.5, which shows that the number of foreshocks is about two decades below the corresponding data for aftershocks. However, this difference of less than two decades is difficult to be identified from Fig.6 because in the vertical range we have many (~ 32) decades. In the revised version we briefly comment this point in the figure caption.

--p13 aftershock location:

"First of all, we observe that most aftershocks occur close to the border of the mainshock's fractured area. This scenario is strongly supported by recent observations of the aftershock organization after big mainshocks 59. Most of the #59 reference earthquakes are subduction zone earthquakes below 15 km depth (see p2). Are the patterns of "universality" classes still stable with depth, is not discussed? Some recent papers point the others way (e.g., Hainzl et al. JGR 2019)''

As discussed at point 1 of this answer we expect that our model can be applied also to subduction zones. In our model we can expect that the parameters θ and also σ can be affected by changes in depth. In particular we find that the aftershock productivity increases with θ and this can be consistent with the findings by Hainzl et al., if θ decreases with the depth. However it is not easy for us to establish how θ changes with depth.

--A modified figure 2 to present the results the same way as cited in #59 reference is more than welcome?''

We thank the reviewer for this suggestion and in the Suppl. Fig.5 we perform the same analysis of Wetzler et al. (now ref.65). More precisely we plot the density of events as function of distance from the border of the fractured area, finding results very similar to those obtained by Wetzler et al.

Answer to Reviewer #2
#####

We thank the reviewer for considering our results very interesting. In the following we answer to his specific remark.

1a) The reviewer writes:

--By the notation in text (one-dimensional indices i and j), and by looking at figure 1, the impression is that one deals with a $2 \times N$ model. The sentence ending with "organized on a rectangular lattice (see Fig. 1)" seems to confirm this interpretation. However, figure 2 shows a square lattice with sides L_x and L_y not introduced in the main text. Only in the Methods this becomes clear. The text should be improved to better describe

the geometry of the model.'

We agree with the referee that the reference to Fig.1 was wrong positioned in the text and this could create some confusion. In the revised version of the manuscript we clarify from the beginning that we deal with a 2d model. We also stress that we present results for a square lattice with sides $L_x=L_y=L$.

1b) The reviewer writes:

``Having settled that the model is composed by two 2d carpets on top of each other, it becomes less clear how this is related to the depth in the crust. The Authors argue that the changing with depth of rheological properties of the crust is the feature that justifies the introduction of both velocity weakening and velocity strengthening. How is this represented in the model? Is it a model of a single fault or of a tectonic region?. The excellent performance of the model invites to think of some alternative interpretation: Is it possible that weakening and velocity strengthening are both taking place at the two sides of a fault (in different measure) at every depth?''

Our model describes a single fault embedded in a medium where afterslip relaxation takes place. It is possible to show (Perfettini and Avouac 2004) that this medium can be equivalently described as a fault zone with finite width, characterized by some brittle creep rheology or as a layer with a velocity strengthening rheology. We adopt the latter description which significantly simplifies the mechanical description of our model. From this point of view, the interpretation suggested by the reviewer of a single layer with different friction laws, on the opposite sides, leads to the same mathematical formulation even if it does not correspond to our geophysical interpretation.

We have improved the introduction of the revised version to better describe our model.

2a). The reviewer writes:

``The 'without any fine-tuning' claim in the abstract should be better explained. In the text it is not obvious if the third parameter ϵ (for dissipation) is exactly set to zero or is kept at an arbitrarily small value. The readers should be able to find all information for reproducing the results.''

The same observation has been raised by reviewer 1. We here report the same answer.

Data of Fig.3 show that for all values of θ and for $\sigma \geq 1$, we always observe a magnitude range where data follows the GR law with a b -value ≤ 1 .

Our interpretation, is that small or large values of θ introduce long crossovers which hide the true asymptotic value $b \leq 1$.

In other words, our conjecture is that if we were able to simulate large system with very small ϵ values, after a transient regime at small magnitudes, we expect that a clear regime with $b \leq 1$ would be observed also for $\theta=0.1$ or $\theta=1$.

Concerning the dependence on ϵ we explicitly discuss this point in the Supl. Materials (Supl.Fig.s 2-4).

We show that ϵ only affects the upper cut-off magnitude.

In the revised version we also give the precise ϵ value and we have

also modified the sentence in the abstract mentioned by the reviewer.

2b) The reviewer writes:

``At page 12 Results do not depend on the specific values of $\tilde{\theta}$ and \tilde{f} , thus we only present them for intermediate values: $\tilde{\theta} = 0.5$, $\tilde{f} = 5$. This should be better justified. The range of variability for the disorder is chosen by the authors and $\sigma = 5$ seems rather at its top than at intermediate range.''

The referee is right, the term intermediate referred only to $\theta = 0.5$ and not to σ . We have corrected this sentence in the revised version.

2c) The reviewer writes:

`` $\theta = 0.5$ gives the better Gutenberg-Richter law in figure 3 and indeed it becomes the value over which the Authors focus in other figures. I do not understand the mismatch in cutoff between the curves for $\theta = 0.5$, $\sigma = 5$ in the inset (orange data) of figure 3 and in its main panel (magenta). They should be the same curve, I believe.''

We thank the referee for pointing to our attention to this inconsistency. In the previous version we erroneously put in the inset a figure with a larger value of ϵ . We fixed this problem in the novel Fig.3. We have also changed the color scale so the same color (magenta) is used for the curve $\theta = 0.5$ and $\sigma = 5$ in the inset and main panel.

3) The reviewer writes:

``At page 9 there is a hint at the BTW model. This seems a bit limited. The introduction should include a broader perspective giving the proper credit to the mechanism of self-organized criticality as the origin of avalanches with power laws.''

We agree with the referee that the original result by Bak-Tang-Wiesenfeld was very important and, as matter of fact, our results can be also framed within the research line opened by their results. In the revised Section ``Comparison with previous spring-block models'' we better stress the relevance of the BTW model.

4) The reviewer writes:

``In the Methods, it is not clear if the model is updated sequentially or in parallel. There would be no difference if the model were Abelian as the BTW model, which is probably requiring a hard mathematical proof. The statement ``The precise synchronization of the updates does not matter'' makes sense only if the model is Abelian.''

We use a sequential updating, an information missed in the previous version and now added to the Method Section of the revised manuscript.

We did also some checks and verified that synthetic catalogs obtained via parallel updating appear very similar to those obtained via sequential rules. We thank the referee for this remark which is an interest aspect to further explore in future studies.

We also thank the reviewer for pointing to our attention other minor points which have been all implemented in the revised version of the manuscript.

Reviewers' Comments:

Reviewer #1:

Remarks to the Author:

review of Ms: NCOMMS-20-01427A

I am very ok the way my previous comments and suggestions are being integrated into the revised version of the manuscript.

I recommend the manuscript to be accepted for Nature-communication publication

some minor points to possibly be clarified are listed below

-p10 bottom: "the BTW model does not have parameters..."

this sentence is awkward. Connectivity and distribution rules are not they equivalent to model input parameters ? please clarify

-p10 bottom: "...size of earthquakes (BTW model) is power law distributed".

Is not the BTW exponent of power law far away from the earthquake one ? please clarify

- p12 top: "better reproduced by by class III models"

typo "by by"....

- figure 1: "The fault plane H is subject to velocity weakening friction μ_h "

i failed to catch the "H label" for the fault plane on figure 1 plot ? at any rate, the fault plane label as grey letters is not clear on this figure. Please clarify

figure 1: "d) As in panel (c) for the whole fault plane."

is it to say that during the 5 time steps no event occurs elsewhere on the fault plane except the ones in the pink dotted box? are not the others 4 sequences somewhere on this fault plane ? or does the (d) figure correspond to the fault plane activity only during a time window that overlaps with the "second" sequence, solely ? Please clarify

Reviewer #2:

Remarks to the Author:

The points have been addressed convincingly by the Authors and I recommend publication of the new version, which is also clearer than the first submission.

We thank both reviewers for recommending our manuscript for publication. In the following we answer to the minor points raised by Reviewer #1

- The referee writes:

"p10 bottom: "the BTW model does not have parameters..."

this sentence is awkward. Connectivity and distribution rules are not they equivalent to model input parameters ? please clarify"

In the revised version we have clarified that, once the network of links is fixed, the model does not present other parameters;

- The referee writes:

"p10 bottom: "...size of earthquakes (BTW model) is power law distributed".

Is not the BTW exponent of power law far away from the earthquake one ? please clarify"

It is explicitly stated that the model presents "a too small non-realistic value of β "

In the revised version we try to make this point more clear;

- The referee writes:

"p12 top: "better reproduced by by class III models"

typo "by" by "by".

We have corrected this typo.

- The referee writes:

"figure 1: "The fault plane H is subject to velocity weakening friction μ_h "

i failed to catch the "H label" for the fault plane on figure 1 plot ? at any rate, the fault plane label as grey letters is not clear on this figure. Please clarify"

We have changed the figure according to the reviewer's suggestion.

- The referee writes:

"figure 1: "d) As in panel (c) for the whole fault plane."

is it to say that during the 5 time steps no event occurs elsewhere on the fault plane except the ones in the pink dotted box? are not the others 4 sequences somewhere on this fault plane ? or does the (d) figure correspond to the fault plane activity only during a time window that overlaps with the "second" sequence, solely ? Please clarify"

The reviewer referred to Fig.2 and as correctly observed, panel (d) corresponds to the time window of the second sequence. We clarify this point in the figure caption.